# The Fate of *Deroceras reticulatum* Following Metaldehyde Poisoning

**DOI:** 10.3390/insects12040344

**Published:** 2021-04-13

**Authors:** Amy Campbell, Neil Audsley, Gordon Port

**Affiliations:** 1School of Natural and Environmental Sciences, Newcastle University, Newcastle upon Tyne NE1 7RU, UK; Gordon.Port@newcastle.ac.uk; 2Fera Science Ltd., Sand Hutton York YO41 1LZ, UK; Neil.Audsley@fera.co.uk

**Keywords:** slug, mollusc, molluscicide, pesticide, chemical control, recovery, pest

## Abstract

**Simple Summary:**

The grey field slug, *Deroceras reticulatum* (Müller, 1774) (Agriolimacidae), is one of the most economically important crop pests and is a particular threat to oil seed rape and winter wheat. Without effective slug control, it is estimated that the loss of yield due to slug damage could equate to over £100 million annually for the UK agricultural sector. The molluscicide metaldehyde is one of the most common active ingredients used in slug pellets across the globe; however, its application presents a high risk of surface water pollution and threatens non-target wildlife. The control of slugs by metaldehyde relies on slugs consuming or being in contact with a pellet long enough to receive a lethal dose; otherwise, a slug may recover from the dehydrating and paralysing effects of the molluscicide. This research explores the effect of different concentrations of metaldehyde on slug survival, paralysis and recovery after contact with metaldehyde, and highlights the prospect of slug paralysis being a major contributing factor to successful slug control.

**Abstract:**

The concentration of a pesticide used in agriculture not only has implications for effectiveness of pest control but may also have significant wider environmental consequences. This research explores the acceptability of metaldehyde slug pellets at different concentrations by *Deroceras reticulatum* (Müller, 1774) (Agriolimacidae), and the changes in the health status of the slug when allowed to recover. The highest metaldehyde concentration (5%) yielded the highest slug mortality; however, it also produced the highest proportion of unpoisoned slugs, suggesting the highest level of pellet rejection. Pellets with 1% metaldehyde were as effective as 3% pellets in paralysing a significant proportion of the population after initial pellet exposure; however, more slugs were able to recover from metaldehyde poisoning at 1% metaldehyde compared with 3%. There was no statistically significant difference between the mortality rate of slugs regardless of metaldehyde concentration, suggesting that a lower concentration of metaldehyde may be as effective as a higher concentration.

## 1. Introduction

The primary control of pest slug populations in agriculture is by chemical means using synthetic pesticides. The most commonly used molluscicides are formulated as a bait to be ingestible, grain-based pellets that contain the active molluscicidal agent (wettable powders and sprays are also used for slug control, albeit less frequently). Metaldehyde is widely used across the world as the active ingredient in slug pellets and is typically present in the pellet at concentrations of less than 5%.

Metaldehyde is both a dermal irritant and gastric poison when ingested by a mollusc, which exploits a slug’s requirement for high body water content [1]. It is likely that the irritant effect of the pellet is that which the mollusc will first encounter in the field, whereby, upon physical contact with the pellet, the molluscs’ mucus glands become irritated, which leads to an increase of mucus production and, subsequently, dehydration and desiccation [2]. Upon pellet ingestion by a mollusc, metaldehyde rapidly hydrolyses to acetaldehyde in the gut, which elicits further production of excess mucus from the mucus glands and again leads to the mollusc becoming dehydrated and desiccated [3]. At high concentrations, it is likely that metaldehyde is a nerve poison in molluscs, which may lead to immobilisation on the soil surface and may make slugs susceptible to secondary mortality.

Metaldehyde is successful as a molluscicide, as it causes slugs to lose large amounts of body water via increasing the amount of mucus produced. It is possible that dehydration from metaldehyde ingestion is not sufficient to kill a slug outright, and therefore, it is likely that secondary mortality factors play a large role in slug control with molluscicides.

Once a slug has been poisoned by metaldehyde, it is likely to remain on the surface of the soil surrounded by its own mucus, writhing or completely unable to move due to a lack of body water content, rendering it unable to produce further mucus to propel itself in any direction. The mucus produced by terrestrial gastropods is up to 98% water [4], and therefore, body water loss due to slug locomotion must be compensated by water uptake from the animals’ immediate environment. Gastropod activity has been observed to decrease when body water content drops below 17% of the animal’s normal value of 80% water content [5]; however, lack of movement after metaldehyde poisoning may also be due to paralysis of the muscle mass [5,6]. It is at this point, once paralysed after pellet contact, that a slug is likely to succumb to predation by birds, hedgehogs or ground beetles. Slugs are generally nocturnal and therefore, it is likely that dehydration from encounters with metaldehyde pellets occur during the cooler, damper night time, and therefore, if a slug is still present and paralysed on the soil surface by the morning, the increase in ambient temperature may cause the further dehydration required to kill a slug.

One of the important issues with control of slugs using metaldehyde-based pellets is that although pellets may be successful in dehydrating a slug, the slugs may recover. If there is sufficient soil moisture to allow the slug to rehydrate, they may recover from metaldehyde poisoning. Similarly, if rainfall occurs, rehydration may be facilitated. In previous feeding experiments, it has been suggested that 70% of slugs may recover from consuming metaldehyde [7] when placed in ideal conditions.

Metaldehyde presents a large water pollution risk due to its mobility in soil once leached from slug pellets on agricultural land and was first detected in surface water in the UK in 2007 [8]. Spikes in drinking water pollution levels cost the UK water industry millions of pounds each year, and removal of metaldehyde from drinking water is extremely challenging. Metaldehyde is classed as “semi-persistent” in the aquatic environment and does not respond to conventional drinking water treatment processes [1]. There are a number of developing treatment methods for the remediation of metaldehyde from drinking water systems [9,10,11]; however, these treatments are expensive to both manufacture and implement on a large scale. The pollution risk, coupled with the likelihood that metaldehyde may be less efficient at controlling slug populations due to slug recovery, means that further research into the short-term effects of metaldehyde on slug health immediately after the pest–pellet interaction have occurred, is essential in order to gain insight into the rates of both slug morbidity and paralysis after exposure to metaldehyde. It may be possible to reduce the concentration of metaldehyde in slug pellets if mortality is seen at lower concentrations as well as higher concentrations of metaldehyde, while remaining aware that secondary mortality as a consequence of soil surface slug paralysis is likely to play a large role in slug control with metaldehyde.

This research aimed to assess the likelihood of recovery by *Deroceras reticulatum* (Müller, 1774) (Agriolimacidae) when placed in ideal conditions allowing for rehydration after exposure to metaldehyde pellets, and how the health and recovery status of *D. reticulatum* differed after exposure to various concentrations of metaldehyde.

Specifically, the research objectives were to:Assess the mortality rate of *D. reticulatum* after 14 h of exposure to a metaldehyde pellet of either 1%, 3% or 5% concentration, a commercial metaldehyde pellet and a non-toxic control pellet. It was hypothesised that higher metaldehyde concentrations would result in higher slug mortality.Assess the likelihood of slug recovery after exposure to slug pellets of various metaldehyde concentrations. It was hypothesised that slugs exposed to higher concentrations of metaldehyde would be more likely to die and less likely to recover from metaldehyde poisoning; however, slugs exposed to lower concentrations would be more likely to recover due to the decreased chance of consumption of a lethal dose.Compare the levels of slug paralysis after exposure and during recovery from metaldehyde pellets of various concentrations. It was hypothesised that slugs would be more likely to be paralysed (as opposed to dead) after exposure to lower metaldehyde concentrations (1%) when compared with higher concentrations (5%).

## 2. Materials and Methods

Grey field slugs (*Deroceras reticulatum*) were collected from a public green space in Northumberland, UK. Slugs were selected at random; however, only slugs regarded as healthy were used in the trial. A healthy slug showed no signs of white or excess mucus production, showed responsive movement or skin rippling to light touch and had no obvious deformations. Slug collection took place on the evening before the start of the trial and slugs were stored in damp, lidded containers in a refrigerated environment (3 °C) until transfer into the lab. Slugs were starved and acclimatised for 24 h to a controlled-temperature room, where all replicates took place. Acclimatisation involved storing slugs in a clear Perspex-lidded container on a layer of damped sandy clay loam soil. Immediately before introduction into the experimental arenas, slugs were weighed in order to ensure an even distribution of sizes across treatments (average slug weight: 0.5 ± 0.1 g).

Treatments involved manufactured Lonza Axcela containing 3% metaldehyde and Lonza Axcela non-toxic control pellets, as well as lab-made pellets. Lab-made pellets were made with a base of plain flour mixed with 99.9% pure metaldehyde powder in order to form 1%, 3% and 5% concentrations (*w*/*w*) made to 20 g in weight. Olive oil was then added to the dry mix by 1 mL pipette drops (up to 15 mL depending on the metaldehyde concentration) in order to form a firm but malleable dough. The dough was then cut into 1 g segments and rolled in order to form a pellet.

Commercial slug pellets tend to be lighter than the 1 g lab pellets used in the trial; however, the lab-made pellets were of similar size to the Axcela pellets used in this research and therefore were considered appropriate. Pellets created for use in replicates were made within 24 h of use in the trial and stored in a sealed container in a refrigerated environment until use. Six treatments were tested during the trial: 1%, 3% and 5% lab-made metaldehyde pellets, 3% Lonza Axcela, a non-toxic control pellet and a no-pellet treatment.

*Deroceras reticulatum* were exposed to a pellet type for 14 h, and each slug was only exposed to one type of treatment pellet. Exposure took place in an open-topped plastic arena (56 × 436 × 23 cm) on a layer of damped agricultural clay loam soil approximately 10 cm thick. Soil in the arena was raked in order to produce a fine, even tilth in order to reduce the number of soil clods and shelters for slugs. Around the top of the arena, a barrier of polytetrafluorethylene (Fluon) was painted in order to prevent slugs leaving the arena. Slug behaviour was recorded using Brinno TLC200 HDR Lab-Cam Time Lapse recording cameras (Brinno, Taipei, Taiwan) set at a frame rate of 10 frames per second. Treatment pellets were placed at 9 baiting points evenly distributed around the arena. Following acclimation, slugs were placed at equal intervals around the pellets (a maximum of 6 slugs per trial and 5 replicates per treatment pellet) and left for 14 h, starting at 16:00 p.m., to move freely around the arena and interact with the treatment pellets. Each trial lasted a total of 14 h, with 10 h being in dark overnight conditions and 4 h in the light (2 h at the beginning and 2 h at the end of the trial). Arenas were kept in a controlled-temperature room (15 ± 1 °C) lit with white lights between 06:00 a.m. and 18:00 p.m. and by infra-red lights from 18:00 p.m. to 06:00 a.m. in order to encourage natural day/night slug activity. At any one time, 3 replicates were run simultaneously in separate arenas. A total of 30 slugs were exposed to each treatment pellet type. Video recordings were analysed in order to confirm interaction with treatment pellets. During video analysis, slugs were numbered, and their behaviour recorded throughout the trial.

After the pellet exposure phase of the trial, the health status of each slug was assessed. Slug health status was categorised as follows:Not poisoned—no signs of paralysis and poisoning, the skin rippled to touch and the slug was moving freely, with no excess of mucus noted.Partial paralysis—the slug was on soil surface and moving, but at a slower pace than expected; skin still rippled to touch, and noticeable mucus secretions around the slug.Severely paralysed—the slug was on soil surface and was barely moving, there may have been a large amount of white or clear mucus surrounding the slug, which was likely to be on its side. Skin moved to touch when examined with magnification.Dead—no signs of life; skin did not ripple on touch.

Each slug was then moved into an individually labelled Petri dish lined on the bottom with a round of damp filter paper. Petri dishes were stacked and stored in bags in the controlled-temperature room in order to maintain day–night cycles and regulate temperature. Each slug was checked every day for the following 3 days and a record of their health status was made. Slugs that showed signs of severe paralysis or death were checked with magnification for ripples on the mantle upon touch.

Results were analysed using chi-squared contingency table analysis in order to compare categorical variables in R (R Core Team, Vienna, Austria, 2020) [12]. For the purpose of this research, only 1 control treatment (control pellets) was used in the analysis in order to compare only pellet-containing treatments.

## 3. Results

After 14 h of exposure to treatment pellets of various metaldehyde contents, the health status of *D. reticulatum* showed great variation depending on which pellet it had been exposed to. Slugs subject to the non-toxic control pellet and those that had not been exposed to any pellets showed high levels of good health (i.e., not paralysed), with only one slug observed to have died in the “no pellet” treatment (Figure 1). Slugs from populations that were exposed to pellets containing metaldehyde displayed poorer health, with the majority of slugs showing signs of either moderate or severe paralysis. After 14 h of exposure to pellets, the highest incidence of deaths occurred in slugs that had been subject to 5% metaldehyde pellets (23% death rate) compared with 1%, 3% and Lonza Axcela death rates (7%, 10% and 10% respectively). Slugs showing no signs of poisoning were more than twice as numerous in the 5% metaldehyde treatment population compared with both 1% metaldehyde and Lonza Axcela pellets.

After 72 h in recovery conditions, allowing slugs to rehydrate, slug populations in all metaldehyde treatments suffered approximately 30% mortality, with the remainder of the populations either paralysed or apparently recovered from the effects of the metaldehyde pellet. A higher concentration of metaldehyde in the pellet did not yield a higher mortality rate, although the number of both severely and moderately paralysed slugs was lower in 5% metaldehyde compared with the 1%, 3% and Axcela treatments (Figure 2). The relative mortality between treatments did not change from 14 h to 72 h (X^2^_(16)_ = 4.918, *p* > 0.05). Slugs exposed to metaldehyde pellets recovered in all treatments, and recovery was slightly higher in both 1% and 5% metaldehyde pellets (20% and 23% recovery) when compared with both of the 3% metaldehyde pellet treatments (10% and 13% recovery rate in 3% lab-made and Axcela pellets, respectively), though not significantly so.

A proportion of slugs remained paralysed in all metaldehyde treatments after 72 h of recovery; however, there was a higher level of paralysis in lower-concentration treatments when compared with 5% metaldehyde. The results of chi-squared contingency analysis showed that there was no significant interaction between treatment and the number of slugs paralysed (moderately and severely combined) after initial exposure and 72 h after recovery (X^2^(4) = 1.784, *p* > 0.05). The commercial treatment, Lonza’s Axcela pellet, showed a similar proportion of paralysed slug population to that of 1% metaldehyde treatment but had high proportions of severely paralysed slugs when compared with lab-made treatments after 72 h of recovery.

Paralysis proportions across all metaldehyde treatments decreased after recovery compared with proportions initially after exposure. Figure 3 shows the outcomes after 72 h of initial moderate or severe paralysis for each of the metaldehyde treatments. Slugs categorised as moderately paralysed tended to stay in that category or recover after 72 h, whereas slugs categorised as severely paralysed stayed in that category or died.

After 14 h of exposure to metaldehyde, the majority of slugs from each treatment were assessed to be suffering from either severe or moderate paralysis (Figure 3). For the commercial pellet, Axcela 3% metaldehyde, there were twice as many slugs assessed to be severely paralysed compared with moderately paralysed after initial exposure. Almost half of the Axcela treatment slugs that were moderately paralysed then recovered from the poisoning effects of metaldehyde after 72 h of recovery, whereas half of the slugs severely paralysed by Axcela did not recover and remained severely paralysed, with approximately 30% of severely paralysed slugs dying by the end of the recovery period. The fate of moderately paralysed slugs observed in other metaldehyde treatments generally followed the same outcome as the Axcela treatment, with slugs in this category more likely to recover rather than die from exposure to metaldehyde after 72 h of recovery. Severely paralysed slugs in the lab-made metaldehyde treatments with 3% and 5% metaldehyde were 50% likely to succumb to metaldehyde and die during the recovery period; however, slugs severely paralysed by the lowest (1%) metaldehyde pellet were 50% likely to remain severely paralysed after the recovery period, with a small percentage (>15%) of slugs recovering slightly and showing signs of moderate paralysis.

## 4. Discussion

This research aimed to provide insight into the effect of metaldehyde on the health of *D. reticulatum* and how the rate of recovery of a slug poisoned by consumption of molluscicidal bait and placed in ideal conditions may change with the metaldehyde concentration of the bait.

In this lab-based research, *D. reticulatum* were exposed to slug pellets of various metaldehyde concentrations over a 14-h period in order to replicate one full night of foraging in the field. The health status of the slugs was then assessed in terms of mortality and paralysis. It was hypothesised that increasing the concentration of metaldehyde would have one of three outcomes for the slug behaviourally:The slug would detect the higher concentration of metaldehyde and reject the pellet entirely.The slug would feed on the pellet for a short time, causing the slug to be unaffected or moderately paralysed.The slug would feed on the pellet and consume a lethal dose of metaldehyde more frequently than with lower concentration pellets.

Paralysis status was of interest in this study, given the evidence that slugs often succumb to secondary mortality in the field after metaldehyde poisoning [13,14]. 

The treatment groups of *D. reticulatum* that were exposed to the non-toxic control pellets and those that were exposed to no pellets, showed, as expected, good health both after initial exposure and after 3 days of being in recovery conditions. This is unsurprising due to the lack of molluscicide present in the replicates but confirms that for the non-toxic control pellet (which was identical to the Lonza Axcela pellet but without metaldehyde) it is the metaldehyde in the pellet that had the molluscicidal effects [15]. 

The 1%, 3% and Lonza Axcela pellet treatment groups had low levels of slugs seemingly unaffected by the pellets (approximately 10% for each group). Over 20% of the slugs exposed to 5% metaldehyde were unaffected by the pellets both after initial exposure and 3 days of recovery. This could suggest that the 5% metaldehyde pellet is more easily detected by a *D. reticulatum* and is more likely to be rejected when encountered in the field, which would support Hypotheses 1 and 2 mentioned previously. 

Paralysis of *D. reticulatum* was divided into two categories: moderately or severely paralysed. It was predicted that moderately paralysed slugs would consume less molluscicide (weight for weight) compared with severely paralysed slugs and therefore would be more likely to recover in the field. On the other hand, severely paralysed slugs were more likely to not recover or be predated upon due to their inability to move. After initial pellet exposure, groups from all treatments had a high proportion of slugs that were moderately or severely paralysed. More than 80% of slugs in the 1% treatment group showed signs of paralysis after initial exposure, which indicates that a large proportion of slugs accepted the low-concentration slug pellet and consumed enough of the pellet in order for the molluscicide to begin to have detrimental effects on the slug’s health. In an agricultural setting, it could be that this would be sufficient for predation or daytime dehydration to occur within the paralysed group, removing a large proportion of the pest population. A similar average weight of slugs between each replicate was selected in order to reduce age bias and to reduce the risk of distorting the results between treatments based on slug size; for example, heavier slugs are likely to be larger in physical size and therefore capable of consuming a larger quantity of pellets. It is also likely that larger slugs require a larger lethal dosage of metaldehyde.

Both the 3% and Lonza Axcela pellets yielded slightly higher levels of severely paralysed slugs when compared with the 1% and 5% metaldehyde treatment groups after initial pellet exposure. In the field, this could mean that more slugs would be lying on the soil surface and would be likely to succumb to secondary mortality. Fewer slugs were paralysed after initial exposure to 5% metaldehyde, supporting the hypothesis that this high concentration of pellet could be seen to work on an “all or nothing” basis depending on individual slug behaviour.

After 3 days of recovery, an extremely small proportion of slugs remained paralysed after exposure to 5% metaldehyde, and the other individuals had either died or recovered from the poisoning. Levels of paralysis remained relatively high (around 40%) for 1%, 3% and Axcela pellets, which implies that recovery from metaldehyde poisoning may take a long time and that other factors, such as food and nutrient availability, may play a role in recovery. Remaining in a paralysed state for longer increases the risk of slugs being predated upon or dehydrating due to weather conditions, and it could be that this is as effective a control by metaldehyde as causing direct slug mortality. Moderate paralysis was seen in higher proportions after 3 days of recovery for both the 1% and 3% metaldehyde groups, which suggests that slugs can survive for longer if only moderately paralysed, which again has implications both for secondary mortality and recovery. 

The highest-concentration metaldehyde treatment group (5%) gave rise to the highest proportion of deaths after initial exposure (approximately 20% of the population). This is still a low proportion of a population and is unlikely to translate to noticeable protection of crops, but our experiments were in very artificial conditions. Lower concentrations of metaldehyde resulted in approximately 10% mortality of the treatment groups after initial contact. Low mortality rates imply that the slugs had not consumed a sufficient amount of the pellet which would deliver a lethal dose. If a slug is unlikely to consume the lethal dose of a pellet over one night of foraging, this may be problematic for growers if there is rainfall shortly after application. Metaldehyde is highly mobile and is easily leached out of a pellet after rainfall and on contact with soil moisture [16,17,18], and therefore, pellets may lose their potency quickly, depending on environmental factors. Ideally, pellet consumption should deliver the lethal dose in the first meal or certainly within that night of foraging in order to avoid issues with metaldehyde loss. The proportion of slugs that died across all metaldehyde treatments after 3 days of recovery was around 30% and this is similar to previous metaldehyde recovery studies [19]. Mortality is still lower than required in order to produce effective slug control in the field and may be the reason for growers reporting slug-related crop damage despite employing molluscicide pellets to control the population [20].

Recovery rates for both the 1% and 5% treatment groups were approximately 20% of the poisoned population. If almost a fifth of the population can recover from the dehydrating effects of metaldehyde after 3 days of rehydration, this indicates that a substantial proportion of the population may be sustained after exposure to metaldehyde. For both the 3% and Axcela pellets, only 10% of the population recovered, meaning more of the populations exposed to 3% metaldehyde had either suffered mortality or remained paralysed. Ideal recovery conditions are difficult to simulate in a lab-based setting due to the many abiotic and biotic factors that could have a positive or negative effect on slug health. Rehydration is likely to be the key factor in determining whether a slug may recover from metaldehyde poisoning [21], and it is for this reason that water availability was the only factor included in the recovery setting. *D. reticulatum* is able to survive for many days without regular feeding [4] and therefore, it was predicted that a lack of food would not have a significant influence on the ability of the slug to recover. Slugs from all replicates were obtained from the same location, and it could be that there were environmental factors that may have influenced the fitness success, genetics or phenology of the population, as well as the potential risk of previous metaldehyde exposure that may have influenced the results in this research. Using multiple sites for the acquisition of slugs may lessen the risk of site-specific influences on behaviour and genetics. Furthermore, though no such interactions were recorded during data collection, it should be considered that slugs may produce alarm pheromones in the early stages of metaldehyde ingestion, which may influence the behaviour of other slugs around metaldehyde pellets.

Metaldehyde slug pellets are sold commercially at concentrations ranging from 1% to 5% metaldehyde, and the industry standard for agronomic uses tends to be around 3% concentration. While agricultural slug pellet application in England and Wales has remained relatively constant over the past 10 years, there has not been a marked decrease in the amount of slug damage to crops at the end of each growing season, as would be expected with successful pesticide use [22]. Metaldehyde pellets are of major concern to the UK water industry due to the chemical’s high mobility in the environment, particularly after heavy rainfall. Metaldehyde is a pollution risk to surface water and it can be detected in river catchments, where concentrations regularly exceed the EU Drinking Water Directive limit of 0.1 μg/L of a single pesticide [1]. On top of the pollution risk, metaldehyde is extremely difficult to remove from drinking water systems once detected and has the potential to cost water bodies millions of pounds in setting up effective treatment systems [23]. Metaldehyde pellets pose a risk to non-target organisms, particularly animals found in similar habitats to slugs or those that feed on invertebrates, such as hedgehogs and birds [24,25,26,27]. 

Considering the risk to the environment, it is imperative that the control of slugs is made as efficient and as effective as possible. Observations of continued slug damage at the end of the growing season indicate that there may be issues at the pest–pellet interaction level that are inhibiting the pellet from working as an effective molluscicide [28]. 

## 5. Conclusions

Although consumption of 5% metaldehyde resulted in the highest mortality rate after initial exposure, there was also a higher proportion of slugs that had not been affected by the pellet. It is likely that a high-concentration pellet results in higher levels of molluscicide detection by the slug and is therefore more likely to be rejected by a high proportion of a population in the field. Both the commercial and lab-made 3% pellet produced similar results in terms of low initial mortality and high levels of paralysis, although after recovery, it was noticeable that a higher proportion of slugs exposed to the commercial pellet remained severely paralysed and would be arrested on the soil surface when compared with the non-commercial 3% pellet, which implies that pellet formulation is key in producing an effective pellet. Exposure to 1% pellets yielded interesting results in that the proportions of slugs paralysed, dead and recovered were not dissimilar to those exposed to 3% pellets. 

While 3% pellets are generally observed as the industry standard, research into the effectiveness and stability of a 1% pellet in an agricultural setting that could yield similar mortality and paralysis as 3% pellets may be a way of reducing the amount of metaldehyde applied to the environment. Maintaining pellet potency would be essential in designing a 1% pellet, as current formulations allow metaldehyde to leach into surface water, and it would be vital that the pellet retained the molluscicide for as long as possible due to its reduced concentration. Paralysis by metaldehyde pellets may be as important as direct mortality due to the existence of natural predators; however, the rate of predation is extremely difficult to quantify [13]. Consumption by predators of slugs that have been paralysed with 1% metaldehyde may be less likely to cause secondary poisoning in the food chain and therefore, the replacement of 3% pellets with 1% pellets may be beneficial not only to the water industry but to non-target organisms. 

## Figures and Tables

**Figure 1 insects-12-00344-f001:**
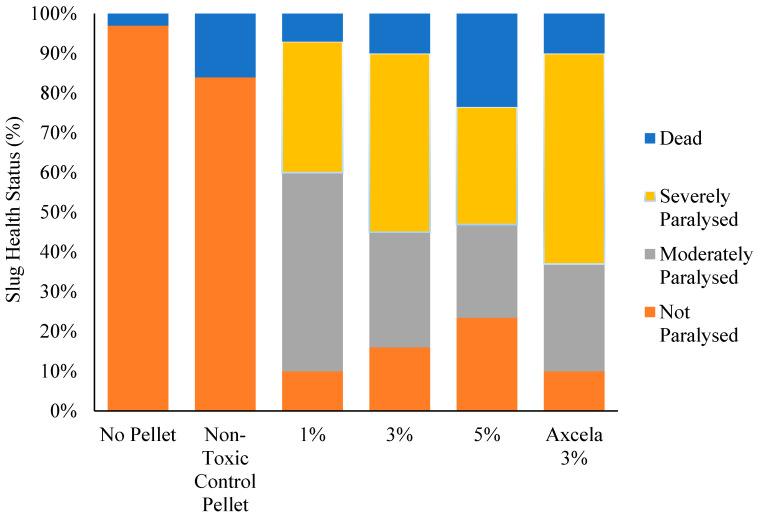
Health status of a population of 30 *Deroceras reticulatum* after 14 h of exposure to pellets of varying levels of metaldehyde concentration. Each slug population was subject to only one treatment pellet (*n* = 30)**.** The relative mortality of slugs between treatments did not change from 14 h to 72 h (X^2^_(16)_ = 4.918, *p* > 0.05).

**Figure 2 insects-12-00344-f002:**
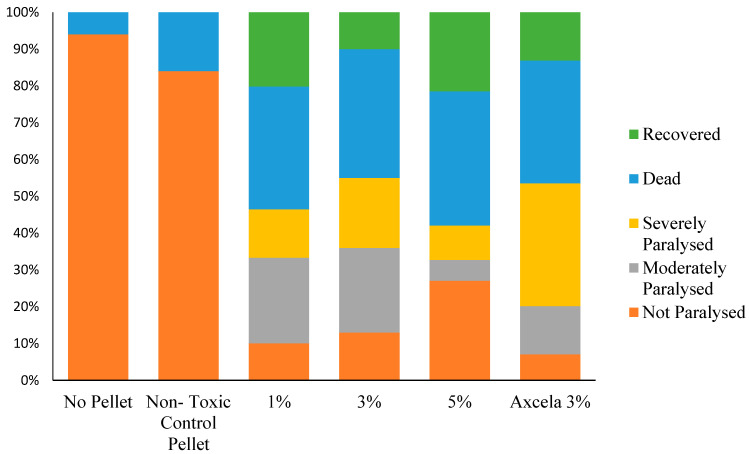
Health status of populations of 30 *D. reticulatum* after 72 h in a recovery environment after exposure to slug pellets of various metaldehyde content (*n* = 30). There was no significant interaction between treatment and the number of slugs paralysed (moderately and severely combined) after initial exposure and 72 h after recovery (X^2^_(4)_ = 1.784, *p* > 0.05).

**Figure 3 insects-12-00344-f003:**
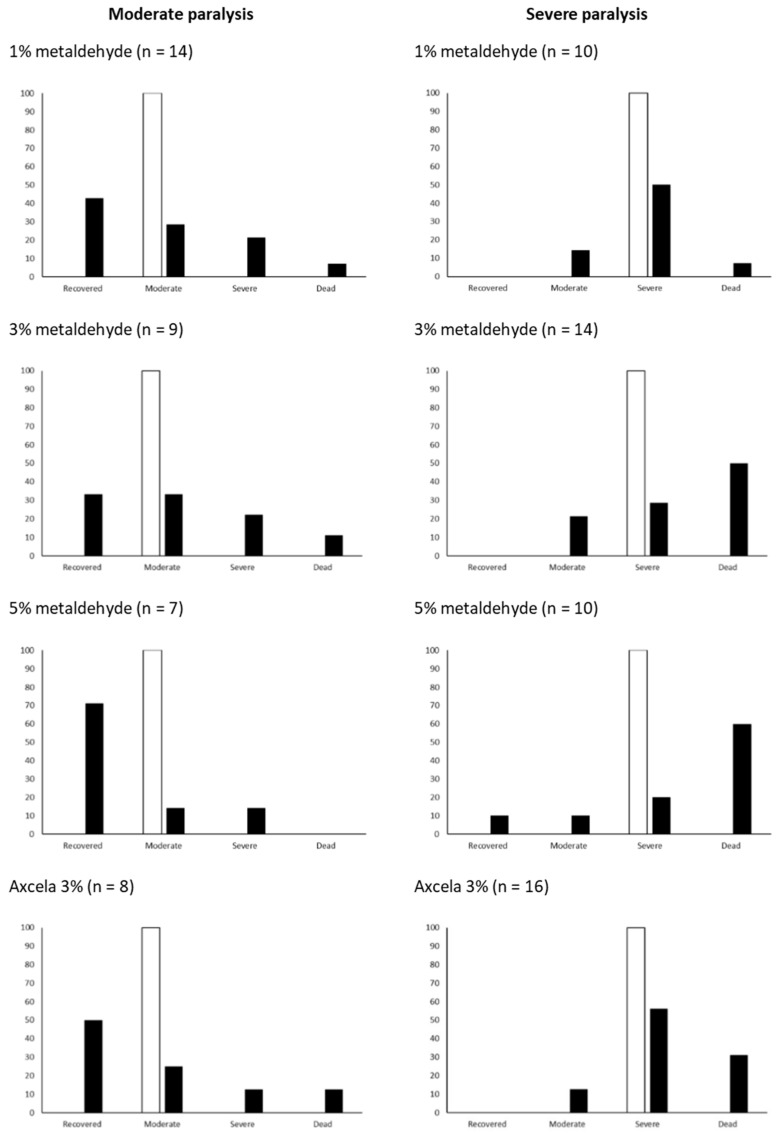
The outcomes of moderate or severe poisoning with metaldehyde baits. Baits were experimental with 1%, 3% or 5% metaldehyde or commercial Axcela (3%) pellets. Slugs were assessed at 14 h after the start of exposure (white bars). The percentage of poisoned slugs that were assessed in the same or different category are shown at 72 h as black bars. The numbers of slugs in each cohort (indicated above each figure) have been converted to percentages for comparison purposes.

## Data Availability

The data represented in this study are available on request from the corresponding author.

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
