# Peer review of "The Fate of *Deroceras reticulatum* Following Metaldehyde Poisoning"

_insects, 2021, doi:10.3390/insects12040344_

Round 1

Reviewer 1 Report

Deroceras reticulatum is one of the most common and dangerous pests in Europe. Control pests is one of the major problems in agriculture. Therefore, the importance and timeliness of the peer-reviewed study is undeniable.

Reducing the release of pesticides into the environment, in particular metaldehyde, is necessary to reduce the negative impact on living organisms of agrocenoses. Therefore, the three problems posed at the end of the Introduction of the article need to be investigated.

The Material and methods are written in detail, which makes it possible to repeat the research of the authors to any interested researcher. However, there are technical disadvantages in the article.

  1. The authors assessed the condition of the slugs only externally. It would also be interesting to assess the presence of live nematodes and acarines moving near the respiratory opening in the mucus from the integuments of animals. These parasites make it possible to indirectly assess the physiological state of an individual slug before and after the experiment.
  2. The Keywords of the article should not duplicate the words already given in the title of the article.
  3. Recording of time and temperature is incorrect (for example, lines 146, 150). 
  4. The Results should not contain separate sentences describing the research process. These sentences should be removed or moved to the Material and methods. 
  5. The percent symbol (%) must be indicated next to each of the digits, as is done on line 188, but not on line 100.
  6. Table 1 is not informative. I recommend to present it in the form of text.
  7. The gram (g) must be separated from the digit with a space, for example, lines 127, 129, 131 and others.
  8. Figure 3 needs to be remade following the pattern of Figures 1 and 2. It is untidy, incomprehensible and poorly understood by the readers. If the authors of the article consider it impossible, the data should be presented in the form of a table.
  9. It is better to indicate the results of statistical processing using the chi-square test (for example, lines 201, 214) in the figure or in its title, and not in the text of the article. So the reader will immediately see whether the differences are reliable, and will not look for this fact in the text.
  10. The first mention of a species both in the Abstract and in the text of the article must be accompanied by the name of the author who described the species, the year of description and in brackets by the detachment and family.
  11. The repeated mention of the species in the article must be accompanied by a reduction of the genus to one letter (for example, lines 92, 94, 260, 273 and many others).
  12. In the list of references, it is not necessary to write every word in the title of the article with a capital letter.
  13. There are numerous inaccuracies in the literature, such as lines 425, 442, 443, 451.
  14. It is better to write a parenthesis after the number and a semicolon after the text (lines 265–269) in a numbered list. If this list contains a continuing clause after the colon (lines 98–111), the requirements are the same. The list can be made with a capital letter (for example, this text with comments), but then the sentence before it must be complete and must not end with a colon.
  15. After the mention of the equipment or computer program in the text, the manufacturer, country and year must be indicated in brackets (for example, lines 142, 175).
  16. Axcela® pellets must be accompanied by the ® symbol everywhere (for example, lines 190, 204 and many others).
  17. UK should be written consistently (for example, lines 12 and 114).
  18. Species names, for example Oil Seed Rape and Winter Wheat, do not need to be capitalized (line 10 and others). These technical disadvantages do not diminish the scientific and practical value of the research. After eliminating them, I recommend the article for publication.

Reviewer 2 Report

The paper describes a topic of great interest to the arable crops industry in Europe.

In general the paper is well written and appropriately structured, albeit a little repetitive. There is probably room to trim back 10% or more in length with attention to reducing repetition.

The issue of low control levels and high recovery rates being influenced by metaldehyde concentration has long been known and the subject to many experiments and trials, including much unpublished work that has underpinned commercial bait and pellet development. As such the work is not particularly novel. Nonetheless, the work being reported takes a novel, more standardized approach.

There are several questions that are not addressed by the authors.

Firstly, the authors state the slugs were weighed in order to ensure a uniformity in size across treatments. However, they do not actually state the size range range of the slugs used in the trial. Further, they do not discuss the possible influence of slug size on both acceptance of pellets and susceptibility to metaldehyde.

Secondly, the authors have not addressed the possible interaction of slugs within replicates on rates of acceptance/ingestion of pellets, amounts of pellet ingested and rates of poisoning. The video methodology used offers the opportunity to examine if there were any interactions between slugs that might influence emergent treatment outcomes - such as the possibility that an encounter of one pellet by a slug may influence subsequent slug encounters with that pellet or indeed other pellets within the same arena. There has been some suggestion that 'stress/alarm pheromones' or similar types of responses elicited by slugs in the initial stages following ingestion of a metaldehyde-containing pellet by deter subsequent encounter by other slugs.

Because there are always the possibility of interactions among individuals within an arena, then effects on individuals within arenas (i.e. within replicates) are not entirely independent. Thus in this case the appropriate statistical analyses should not be based on 30 slugs per treatment, but on 5 replicates per treatment. In Figure 1 for example, the graphic should be based on the mean response across the 5 replicates, i.e. calculated as the slug health status of the 6 slugs per replicate averaged across the 5 replicates.

Thirdly, the authors report a trial based on a slugs drawn from a single population, but make no comment on the generality of the result. That is, the authors do not consider the possibility that slug populations may vary both in their response to pellets and to metaldehyde with regional- or site-specific environmental legacy effects on phenology, various fitness parameters, genetics, prior exposure to metaldehyde, etc. It would have been be appropriate to replicate the trials across slugs drawn from different populations/locations.

The authors use the term 'trials' and 'trials per treatment'. This should be replicates, and replicates per treatment. 

There are some minor editorial issues throughout, including in the References inconsistence in capitalization of the first letter in words in the article titles.
